# Prevalence of Occult Central Lymph Node Metastasis by Tumor Size in Papillary Thyroid Carcinoma: A Systematic Review and Meta-Analysis

**Liyang Tang \***, **Roy W. Qu** , **Jaimie Park, Alfred A. Simental and Jared C. Inman**

Department of Otolaryngology—Head and Neck Surgery, Loma Linda University Medical Center, Loma Linda, CA 92354, USA; jinman@llu.edu (J.C.I.)
**\*** Correspondence: ltang@llu.edu

**Abstract:** Background: While papillary thyroid carcinoma (PTC) is associated with high occult central neck metastasis (CNM) rates, prophylactic central neck dissection (pCND) is controversial. This meta-analysis aims to look at the occult CNM rate according to tumor size. Methods: A literature search was conducted in PubMed from inception to April 2023. Inclusion criteria were primary studies that determined occult CNM rates in cN0 PTC by tumor size. Heterogeneity, influential case diagnostics, and proportion data were evaluated with Cochran's Q-test, Baujat plots and Forest plots, respectively. Results: Fifty-two studies were included in this meta-analysis. The findings demonstrated an occult CNM rate of 30.3% for tumors $\leq 5$ mm, 32.7% for tumors $\leq 1$ cm, 46.0% for tumors between 1 and 2 cm, 43.1% for tumors between 2 and 4 cm, and 61.2% for tumors > 4 cm. The heterogeneity of each study group was high, though no publication bias was noted. While there was a trend towards increased occult CNM rates with larger tumors, comparisons between different size cutoffs varied in significance. Conclusion: This comprehensive review affirms that occult CNM is high and that an ipsilateral pCND can be justified in all PTC patients for accurate differentiation between Stage I and Stage II disease and its clinical implications.

**Keywords:** papillary thyroid carcinoma; central neck dissection; occult lymph node metastasis

## 1. Introduction

Papillary thyroid carcinoma (PTC) makes up approximately 80–85% of all thyroid malignancies [1]. The frequent presence of lymph node metastases, particularly in the ipsilateral central neck compartment (level VI), is a characteristic feature of PTC, with the incidence ranging from 20% to as high as 90%, depending on the study population and tumor size [2–6]. Despite its high prevalence, the optimal management of clinically node-negative (cN0) patients remains controversial, particularly regarding the role of prophylactic central neck dissection (pCND) [3,7–9].

The debate over the management of occult lymph node metastasis in PTC centers around the risks of understaging the cancer and the potential for increased locoregional recurrence due to undertreatment in forgoing a pCND versus the potential increase in morbidity while performing a pCND [10–12]. pCND has been associated with improvements in staging accuracy and reductions in locoregional recurrence [10–13]. Multiple meta-analyses including those by Wang and Chen et al. suggest that pCND is associated with a reduced risk of locoregional recurrence, although not all studies agree and a retrospective study by Dobrinja et al. did not show such an association [12,14,15]. Furthermore, multiple studies have shown an association of pCND with increased surgical morbidities, including both temporary and permanent hypoparathyroidism and injury to the recurrent laryngeal nerve, which may outweigh the potential benefits of a pCND in cN0 patients [3,12,15]. However, surgeries performed at high-volume centers that carry out careful identification and preservation of the parathyroids and recurrent laryngeal nerve and only performing

ipsilateral pCNDs make permanent morbidities comparable to those of a thyroidectomy alone [16–19].

In light of the ongoing debate, there is a need to further elucidate the rate of occult central neck metastasis (CNM) in PTC and the role of pCND. To our knowledge, despite the wide range of occult central neck metastases reported in the literature, there has not been a summary of the rate of occult central neck metastasis by tumor size. Additionally, with the recent shift toward the de-escalation of care for PTC and the consideration of hemithyroidectomy in PTC of less than 4 cm with no adverse features, it is unclear if patients with smaller tumors are being understaged and undertreated [7]. Under current AJCC 8th edition staging guidelines, lymph node positivity in patients of >55 changes from Stage I to Stage II [19]. The implications of this clinically should not be taken for granted as they usually escalate treatment to include completion thyroidectomy, thyroid-stimulating hormone (TSH) suppression, and a consideration of radioactive iodine [7]. Thus, this systematic review and meta-analysis aims to estimate the rate of occult level VI metastasis in PTC by tumor size.

## 2. Materials and Methods

### 2.1. Search Strategy

Two independent reviewers (LT and JP) conducted a comprehensive literature search using PubMed to identify studies that evaluated the percentage of occult central neck disease in cN0 PTC by tumor size. Any discrepancies were discussed and settled with the third reviewer (JI). The search terms included the following: (papillary thyroid carcinoma) AND thyroidectomy AND ((central neck dissection) OR (level VI neck dissection) OR (level 6 neck dissection)). No restrictions on the publication date or language were applied initially. However, due to language limitations, only articles published in English were included in the final analysis.

### 2.2. Study Selection

After the initial search, duplicate records were removed, and the remaining studies were screened based on their titles and abstracts, as shown in the PRISMA diagram (Figure 1) [20]. Full-text articles were then assessed for eligibility according to the predefined inclusion and exclusion criteria. Inclusion criteria were primary studies that looked at the occult lymph node metastasis rate according to tumor size in cN0 PTC. cN0 was defined as patients with no lymphadenopathy identified via a physical exam or on ultrasonography before surgery. Exclusion criteria included the following: (1) papers that included patients with lymphadenopathy identified either on a pre-operative physical exam or on imaging, (2) studies that included malignancies other than well-differentiated thyroid cancer, and (3) review articles, case reports, and conference abstracts. Discrepancies between the two authors (LT and JP) regarding study eligibility were resolved through discussion and consultation with a third author (JI).

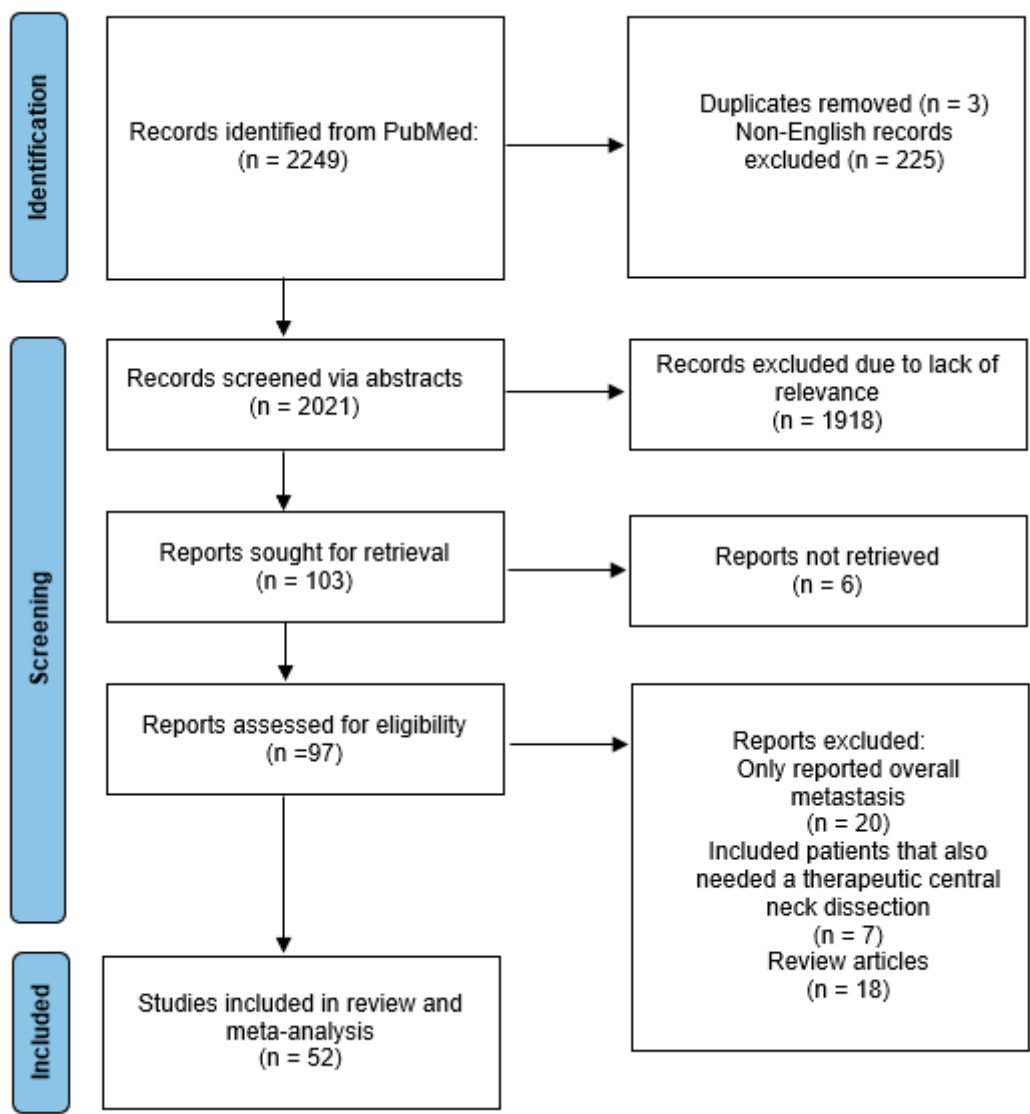

**Figure 1.** PRISMA diagram [20].

*2.3. Data Extraction*

Data extraction was performed independently by the two authors using a standardized data extraction form. The following information was extracted from each study: the first author's name, publication year, country of study, study design, sample size, age range of patients included in the study, tumor size, post-operative pathologic nodal staging, number of central lymph nodes analyzed, rate of occult lymph node metastasis, locoregional recurrence, and average follow up time (Table 1). In papers where only the ipsilateral and/or contralateral CND metastasis rate was reported per tumor size, but the overall CND metastasis rate was not, the ipsilateral CND metastasis rate was extracted and used in the analysis.

**Table 1.** Included studies with tumor size categories that were analyzed.

| Name | Pub Year | Country | Study Design | Age Range (Years) | Occult Metastasis #/Total # (%) | # of LNs Sampled Mean (Range) | Locoregional Recurrence #/Total Pos (%) | Mean Fu Time (Range, Months) |
|---|---|---|---|---|---|---|---|---|
| Wada, N et al. [21] | 2003 | Japan | Retrospective | 17–72 | ≤1 cm: 143/235 (60.9%) | 13.2 | 1/235 (0.43%) | 61.6 (13–144) |
| Lim, YC et al. [22] | 2009 | South Korea | Retrospective | | ≤1 cm: 27/86 (31.4%) | 11.5 (2–19) | | |
| Costa, S et al. [23] | 2009 | Italy | Retrospective | | ≤1 cm: 16/36 (44.4%) | | 8/126 (6.3%) | 47 (0–159) |
| Bonnet, S et al. [24] | 2009 | France | | 18–73 | ≤1 cm: 14/51 (27.5%)<br>1–2 cm: 28/64 (43.8%) | | | >12 |
| Koo, B et al. [25] | 2009 | South Korea | Prospective | | ≤1 cm: 7/28 (25%) | 6.2 (2–23) | 0/28 (0%) | 24.4 (12–38) |
| Moo, T et al. [26] | 2009 | USA | Prospective | | ≤1 cm: 4/16 (25%) | | | |
| So, Y et al. [27] | 2010 | South Korea | Retrospective | | ≤1 cm: 202/551 (36.7%) | | 1% | >36 |
| Vergez, S et al. [28] | 2010 | France | Retrospective | 16–81 | ≤1 cm: 39/82 (47.5%) | | | 62 (14–105) |
| Roh, J et al. [29] | 2011 | South Korea | Prospective | 18–77 | ≤1 cm: 22/85 (25.9%) | 8.8 (1–34) | | 46 (12–64) |
| Teixeria, G et al. [30] | 2011 | Brazil | Retrospective | 16–77 | 2–4 cm: 4/9 (44.4%) | 10.1 (6–20) | | |
| Hyun, S et al. [31] | 2012 | South Korea | Retrospective | | ≤1 cm: 19/65 (29.2%) | | 1/65 (1.5%) | 51.3 |
| Kutler, D et al. [32] | 2012 | USA | Retrospective | 11–80 | ≤1 cm: 10/32 (31.3%) | 9.3 (1–37) | | 24.7 |
| Hartl, D et al. [5] | 2012 | France | Retrospective | 10–81 | ≤1 cm: 17/103 (16.5%)<br>1–2 cm: 23/114 (20.2%)<br>2–4 cm: 9/44 (20.5%) | 11 (1–39) | | (6–204) |
| Lee, K et al. [33] | 2013 | South Korea | Retrospective | 20–73 | ≤1 cm: 32/77 (41.6%) | 7.5 (0–26) | | |
| Raffaelli, M et al. [34] | 2013 | Italy | Retrospective | | 2–4 cm: 0/4 (0%) | | | |
| Wang, Q et al. [35] | 2014 | China | Retrospective | | ≤1 cm: 33/94 (35.1%)<br>1–2 cm: 41/76 (53.9%)<br>2–4 cm: 8/17 (47.1%) | 3 (1–16) | | |
| Monacelli, M et al. [36] | 2014 | Italy | Retrospective | | ≤1 cm: 28/108 (26%) | | | |
| Park, JP et al. [37] | 2014 | South Korea | Retrospective | 24–70 | ≤1 cm: 63/193 (32.6%) | (3–33) | 1/287 (0.3%) | (24–44) |
| Varshney, R et al. [38] | 2014 | Canada | Retrospective | 20–91 | ≤1 cm: 23/170 (13.5%) | | | |
| Zhang, LY et al. [39] | 2015 | China | Retrospective | | ≤1 cm: 73/178 (41%) | | | |
| Mao, LN et al. [40] | 2015 | China | Retrospective | | ≤1 cm: 102/332 (30.7%)<br>1–2 cm: 27/57 (47.4%) | | 1/389 (0.3%) | (12–25.5) |
| Xiang, YY et al. [41] | 2015 | China | Retrospective | 18–83 | ≤1 cm: 159/392 (40.6%) | | | |
| Chen, Q et al. [42] | 2015 | China | Retrospective | 10–78 | ≤1 cm: 55/135 (40.7%) | 5.5 (1–21) | | |

**Table 1.** *Cont.*

| Name | Pub Year | Country | Study Design | Age Range (Years) | Occult Metastasis #/Total # (%) | # of LNs Sampled Mean (Range) | Locoregional Recurrence #/Total Pos (%) | Mean Fu Time (Range, Months) |
|---|---|---|---|---|---|---|---|---|
| Ji, YB et al. [43] | 2016 | South Korea | Retrospective | | ≤1 cm: 78/309 (25.2%) 1–2 cm: 51/120 (42.5%) | | | |
| Kim, SK et al. [3] | 2016 | South Korea | Retrospective | | ≤1 cm: 6069/8282 (73.3%) 1–2 cm: 2194/2668 (82.2%) 2–4 cm: 446/574 (77.8%) >4 cm: 26/45 (57.8%) | 7.2 | 270/11,569 (2.3%) | 62.6 (6–216.6) |
| Xue, S et al. [44] | 2016 | China | Retrospective | 14–85 | ≤1 cm: 379/1059 (35.7%) | 5.5 (1–46) | 18/1555 (1.2%) | (60–120) |
| Suman, P et al. [45] | 2016 | USA | Retrospective | | ≤1 cm: 840/9997 (8.4%) 1–2 cm: 1445/8257 (17.5%) 2–4 cm: 1229/5276 (23.3%) >4 cm: 275/1311 (21%) | 4.4 | | |
| Lee, HS et al. [46] | 2016 | South Korea | Retrospective | 16–80 | <1 cm: 128/651 (19.66%) | 5.8 (1–27) | | |
| Oh, HS et al. [47] | 2017 | South Korea | Retrospective | | ≤1 cm: 786/2329 (33.7%) | | | |
| Goran, M et al. [48] | 2017 | Serbia | Retrospective | | ≤1 cm: 23/111 (20.7%) | | 0 | (12–132) |
| Kim, SK et al. [49] | 2017 | South Korea | Retrospective | 39–57 | ≤1 cm: 14/58 (24.1%) | | | |
| Sessa, N et al. [50] | 2018 | Italy | Prospective | 16–85 | ≤1 cm: 82/186 (44.1%) 2–4 cm: 6/21 (28.6%) | 12.5 (6–33) | | |
| Zhang, Q et al. [51] | 2019 | China | Retrospective | 9–74 | ≤1 cm: 400/1304 (30.7%) | | | |
| Chen, BD et al. [52] | 2019 | China | Retrospective | | ≤1 cm: 71/182 (39%) | | | |
| Zhang, C et al. [53] | 2020 | China | Retrospective | 15–74 | ≤1 cm: 101/553 (18.3%) | | | |
| Xue, S et al. [54] | 2020 | China | Retrospective | | ≤1 cm: 22/49 (44.9%) | | | |
| Feng, JW et al. [55] | 2020 | China | Retrospective | 21–79 | ≤1 cm: 123/371 (33.2%) | 5.3 (3–27) | 14/371 (3.8%) | 47 (8–81) |
| Mukherjee, D et al. [2] | 2020 | India | Prospective | | >4 cm: 9/10 (90%) | 6.4 | | (16–24) |
| Wu, Z et al. [56] | 2021 | China | Retrospective | 13–79 | ≤1 cm: 269/670 (40.1%) 1–2 cm: 119/235 (50.6%) | | | |
| Liu, C et al. [57] | 2021 | China | Retrospective | | ≤1 cm: 236/556 (42.4%) | | 0 | (12–18) |
| Kralik, R et al. [58] | 2021 | Slovakia | Retrospective | | ≤1 cm: 25/128 (19.5%) 1–2 cm: 46/140 (32.9%) | | | (58.4–385.7) |
| Huang, Y et al. [59] | 2021 | China | Retrospective | 18–75 | ≤1 cm: 240/484 (49.6%) | 6.3 (1–27) | | |
| Zhou B et al. [60] | 2021 | China | Retrospective | | ≤1 cm: 45/169 (26.6%) 1–2 cm: 21/43 (48.8%) | | | |

**Table 1.** *Cont.*

| Name | Pub Year | Country | Study Design | Age Range (Years) | Occult Metastasis #/Total # (%) | # of LNs Sampled Mean (Range) | Locoregional Recurrence #/Total Pos (%) | Mean Fu Time (Range, Months) |
|---|---|---|---|---|---|---|---|---|
| Li, R et al. [61] | 2021 | China | Retrospective | 22–72 | ≤1 cm: 76/136 (55.9%) | | | |
| Zhou B et al. [6] | 2021 | China | Retrospective | | ≤1 cm: 81/312 (26.0%)<br>1–2 cm: 36/57 (63.2%) | | | |
| Parvathareddy, SK et al. [62] | 2021 | Saudi Arabia | Retrospective | 18–89 | ≤1 cm: 9/22 (40.9%)<br>2–4 cm: 29/38 (76.3%) | | 125/942 (13.3%) | (12–361) |
| Zhong, X et al. [63] | 2022 | China | Retrospective | 18–77 | ≤1 cm: 103/306 (33%) | | | |
| Muthuvel, R et al. [64] | 2022 | India | Prospective | | ≤1 cm: 37/95 (38.9%) | 4.2 (3–8) | 0 | |
| Kang, SK et al. [65] | 2022 | South Korea | Retrospective | 30–86 | ≤1 cm: 47/130 (36.2%) | 9.4 (1–36) | | |
| Shahriarirad, R et al. [66] | 2022 | Iran | Prospective | | 2–4 cm: 13/15 (86.7%) | 10 | | |
| Hartl, DM et al. [67] | 2023 | France | Retrospective | 18–78 | ≤1 cm: 10/27 (37%)<br>1–2 cm: 67/196 (34.2%)<br>2–4 cm: 36/78 (46.2%)<br>>4 cm: 0/1 (0%) | (0–26) | | (1–293) |
| Kwon, O et al. [68] | 2023 | South Korea | Retrospective | 12–79 | ≤1 cm: 1620/2254 (71.9%)<br>1–2 cm: 402/548 (73.4%)<br>2–4 cm: 77/100 (77%) | 6.7 (0–41) | 67/2902 (2.3%) | 112 (7–192) |

Abbreviations: Pub: publication; LNs: lymph nodes; fu: follow up.

*2.4. Statistical Analysis*

All statistical analyses were performed using the metafor (Version 4.0-0) and meta (Version 6.2-1) packages in R (Version 4.3.0; R Project for Statistical Computing, Indianapolis, IN, USA) [69]. Random effect models using the DerSimonian–Laird estimator were created for raw untransformed proportions. Heterogeneity was reported as low (Higgin's $I^2$ = 0–25%), moderate ($I^2$ = 26–50%), or high ($I^2$ > 50%), along with $\tau^2$ and $\chi^2$ statistics. Formal heterogeneity hypothesis testing was evaluated with Cochran's Q-test. Influential case diagnostics were performed using Baujat plots and leave-one-out testing which were reported in the body of the results but not removed from the analysis [70]. Publication bias was assessed with funnel plots and Egger tests using a mixed-effect meta-regression model [71]. Proportion data were reported in Forest plots with counts of central neck disease, total counts, individual study proportions with 95% confidence intervals (horizontal line), study weights (represented by the size of squares), and overall proportions with the reference line and confidence interval (width of diamond). A significance level of $p$ = 0.05 was used for all analyses.

## 3. Results

*3.1. Subsection*

### 3.1.1. Study Selection

Following the initial search and removal of duplicates and non-English papers, 2021 abstracts were reviewed and 103 full texts were analyzed, as shown in the PRISMA flow diagram (Figure 1) [20]. After a review of the articles, 52 studies were included in the final meta-analysis and all are listed in Table 1 [2,3,5,6,21–68,72].

### 3.1.2. Primary Outcome

To report the proportion of occult central neck disease according to size, study outcomes were organized into five primary tumor size groups: <5 mm, <1 cm, 1–2 cm, 2–4 cm, and >4 cm. A random-effects model was utilized to report pooled proportions of occult central neck metastasis (CNM). For primary tumors measuring less than 5 mm, 3304 out of 6579 cases had occult CNM, which represents a weighted rate of 30.3% (15.7–44.9%, 95% C.I.; Figure 2). There was significant heterogeneity ($I^2$ = 99.3%; $\tau^2$ = 0.078; Q = 1834.91; $p < 0.0001$) in this size group, suggesting the true proportion of occult CNM varies across studies. Influence diagnostics revealed that only the study by Kim et al. ($p$ = 70.4%; 68.9–71.9%, 95% C.I.) was an influential study (Supplemental Figure S1) [3]. Kim et al.'s study was a large single-institution retrospective study representing more than 11,000 cases over a nearly 20-year period [3]. Comparatively, this was a disproportionately large study relative to the remainders of the group. A funnel plot was created which did not suggest publication bias when an Egger test was performed (Supplemental Figure S2, $p$ = 0.943).

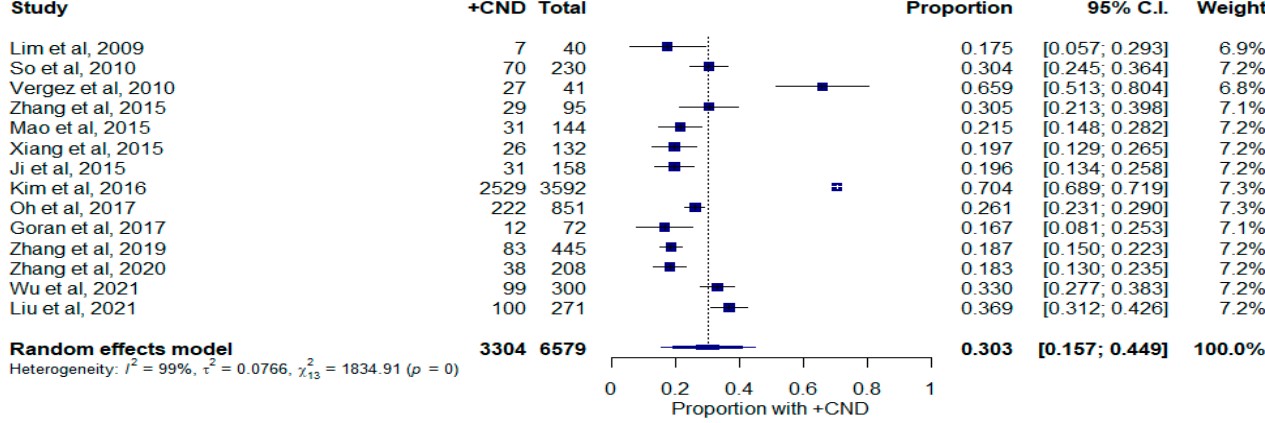

**Figure 2.** Forest plot depicting proportion of occult central neck disease in patients with tumors measuring less than 5 mm. Abbreviations: +CND: positive central neck dissection; C.I.: confidence interval.

The rate of occult CNM using studies with primary tumors measuring less than 1 cm is shown in Figure 3. Studies that reported TNM staging were also included if T1a vs. T1b designations were provided. For primary tumors measuring less than 1 cm, 11,254 out of 29,348 cases had occult CNM, which represents a weighted rate of 32.7% (23.0–42.3%, 95% C.I.). There was significant heterogeneity between these studies ($I^2 = 99.7\%$; $\tau^2 = 0.112$; $Q = 16,296.40$; $p < 0.0001$). Influence diagnostics did not demonstrate a significantly influential study in this size group (Supplemental Figure S1). The funnel plot and Egger test also did not suggest publication bias (Supplemental Figure S2, $p = 0.992$).

The rate of occult CNM in the 1–2 cm group is shown in Figure 4. Similarly, studies that provided T1a vs. T1b distinctions were also included. There were 4536 out of 12,683 cases with occult CNM. This represented a pooled weighted proportion of 46.0% (24.4–67.7%, 95% C.I.). Study heterogeneity was similar in this group compared to that of other tumor sizes ($I^2 = 99.8\%$; $\tau^2 = 0.156$; $Q = 6310.70$; $p < 0.0001$). There were no studies identified on Baujat plots or via leave-one-out analysis (Supplemental Figure S1). There was no statistically significant publication bias detected via the Egger test (Supplemental Figure S2, $p = 0.931$).

For primary tumors measuring between 2 and 4 cm, there were 1828 out of 6159 cases with occult CNM, which equates to a 43.1% (22.0–64.1%, 95% C.I.; Figure 5) pooled weighted proportion. Study heterogeneity was high within this group as well ($I^2 = 99.03\%$; $\tau^2 = 0.342$; $Q = 1034.50$; $p < 0.0001$). Baujat plots and leave-one-out analysis did not demonstrate an influential study (Supplemental Figure S1). The funnel plot and Egger test did not identify any significant publication bias (Supplemental Figure S2, $p = 0.9566$).

Studies that reported tumors measuring > 4 cm are shown in Figure 6. Papers with TNM staging without specified tumor size were not included in this section as it was not possible to delineate T3 tumors according to the inclusion criteria (T3 tumors can indicate a tumor size greater than 4 cm or any size with extrathyroidal extension). There were 1369 cases in total, of which 312 had occult CNM, which represents a pooled proportion of 61.2% (26.1–96.4%, 95% C.I.) There was significant heterogeneity between studies ($I^2 = 94.9\%$; $\tau^2 = 0.123$; $Q = 78.87$; $p < 0.0001$). Mukherjee et al., 2020 ($\hat{p} = 90.0\%$; 71.4–100.0%, 95% C.I.), and Suman et al., 2016 ($\hat{p} = 21.0\%$; 18.8–23.2%, 95% C.I.), had significantly influential studies (Supplemental Figure S1) [2,45]. These studies reported significantly higher and lower rates of occult CNM and did not capture the pooled estimate in their individual confidence intervals while the remainder of the studies did. Suman et al., 2016, conducted a multi-institutional study using the National Cancer Data Base and the rate of occult CNM in this was disproportionately large within this cohort of fewer studies [45]. There was no publication bias identified via the Egger test (Supplemental Figure S2, $p = 0.957$).

Overall, the meta-analysis demonstrated that the rate of occult CNM is high regardless of tumor size. Even for patients with small tumors, occult CNM rates were as high as 30% and 32.7% in patients with tumors smaller than 5 mm and in those with tumors smaller than 1 cm, respectively. There was a trend toward increasing occult CND rates with increasing primary tumor sizes. The occult CND rate peaked at 61.2% in the group with tumors greater than 4 cm.

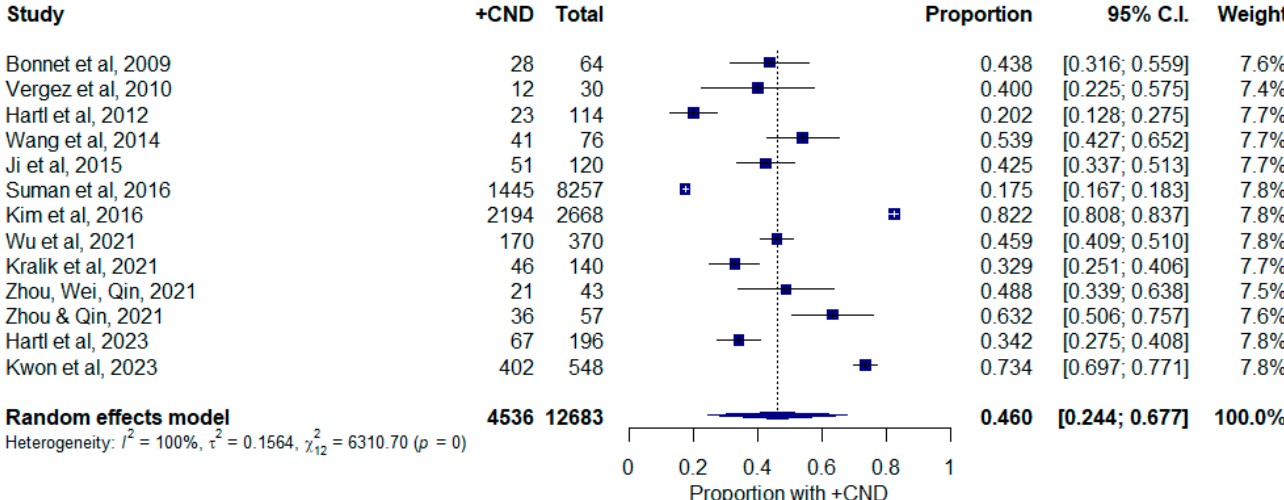

**Figure 3.** Forest plot depicting proportion of occult central neck disease in patients with tumors measuring less than 1 cm. Abbreviations: +CND: positive central neck dissection; C.I.: confidence interval.

**Figure 4.** Forest plot depicting proportion of occult central neck disease in patients with tumors measuring between 1 and 2 cm. Abbreviations: +CND: positive central neck dissection; C.I.: confidence interval.

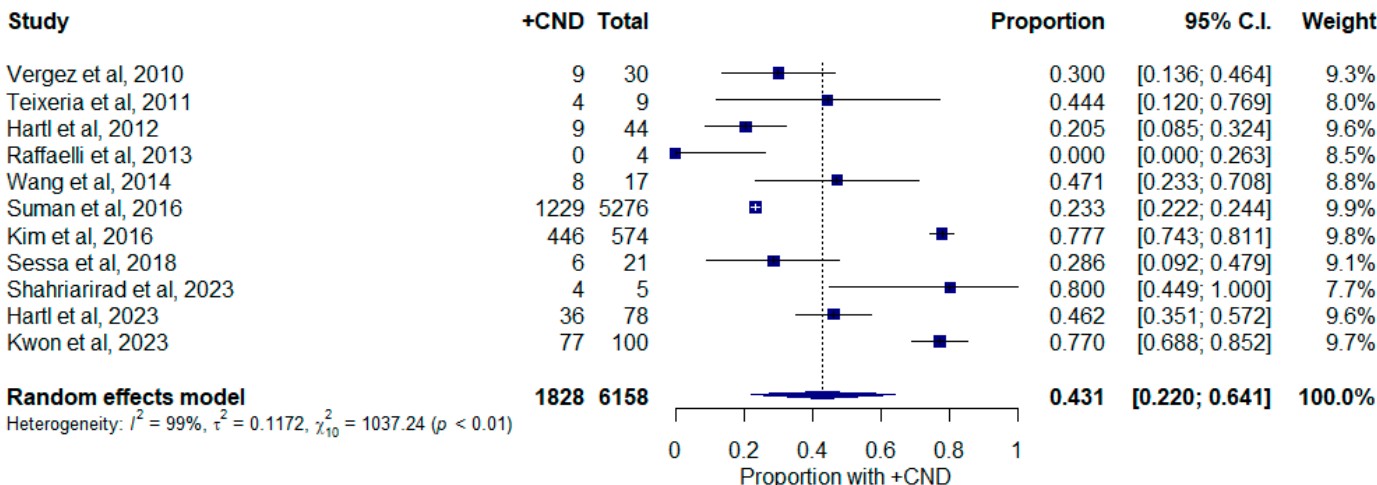

**Figure 5.** Forest plot depicting proportion of occult central neck disease in patients with tumors measuring between 2 and 4 cm. Abbreviations: +CND: positive central neck dissection; C.I.: confidence interval.

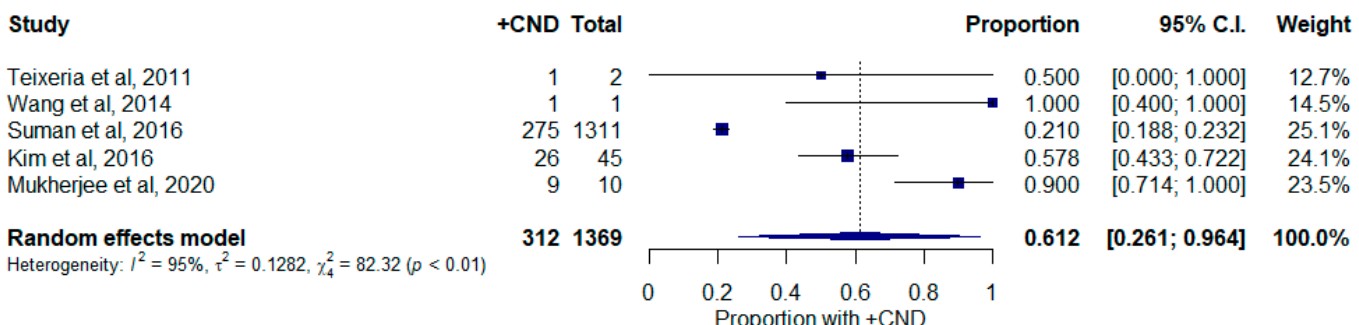

**Figure 6.** Forest plot depicting proportion of occult central neck disease in patient with tumors greater than 4 cm. Abbreviations: +CND: positive central neck dissection; C.I.: confidence interval.

## 4. Discussion

This meta-analysis was designed to investigate the rate of occult CNM in PTC by tumor size to determine if there is a role for pCND for accurate staging in small tumors. The occult metastasis rate was noted to be approximately 30.3% for tumors measuring ≤5 mm, 32.7% for tumors measuring ≤1 cm, 46.0% for tumors measuring 1–2 cm, 43.1% for tumors measuring 2–4 cm, and 61.2% for tumor sizes of >4 cm, demonstrating that even at the papillary microcarcinoma level, the risk of occult metastasis is not negligible.

The current American Thyroid Association (ATA) guidelines recommend the consideration of a lobectomy for PTC of sizes between 1 and 4 cm without adverse features such as extrathyroidal extension or lymphadenopathy [7]. pCND is generally not recommended except in high-t staging or lateral neck dissection [72]. This can potentially lead to the understaging of the disease, as the occult metastasis rate is high, even in papillary microcarcinoma [3,11]. In the eight edition of American Joint Committee on Cancer (AJCC) staging, patients younger than 55 years old are differentiated into Stage I and Stage II by distant metastasis only. However, in patients greater than 55 years old, positive nodal status upstages small intrathyroidal tumors (<4 cm) from I to II [19]. Thus, this meta-analysis suggests that potentially 30–40% of patients with intrathyroidal tumors of sizes less than 4 cm are understaged, which can impact accurate disease-free survival estimates and decisions regarding the use of adjuvant treatments. Furthermore, a national database study using both the National Cancer Database and the SEER database has shown an association between an increased number of metastatic lymph nodes and reduced overall survival rates even in patients younger than 45 years old [73]. Given this finding, the authors of the study

advocated for a revision of the AJCC staging. This further emphasizes the importance of knowing the lymph node status, even in younger patients who are considered to have a lower risk of disease, to allow for more informed decision making.

Certain genetic mutations have also been shown to have a significant negative impact on CNM, locoregional recurrence and survival rates. In particular, the BRAF V600E mutation has been associated with higher rates of CNM, even in patients with cN0 neck disease and micropapillary carcinoma [61,74]. One study by Li et al. even estimated that BRAF V600E increased the risk of CNM by 3.84 times in patients with cN0 necks [61]. Furthermore, BRAF V600E mutation is independently associated with an increased risk of recurrence, which is further compounded by the presence of CNM [75,76]. Another mutation in the telomerase reverse transcriptase (TERT) promotor is also associated with tumor recurrence and decreased survival rates, although there is controversy over whether or not the TERT mutation is associated with increased CNM [77,78]. The presence of both the BRAF V600E and TERT mutations is synergistic and substantially increases the risk of lymph node metastasis, recurrence, and disease-specific mortality rates. In fact, in a study by Liu et al., PTC mortality increased from 2.4% for patients with the BRAF V600E mutation and 6.3% for those with the TERT mutation to 22.7% for those with both mutations [78,79]. Hence, pCND should be strongly considered and perhaps even recommended for PTC with the BRAF V600E mutation and especially in those with both these mutations, as central lymph node status can influence the decision to pursue further adjuvant therapies.

CNM is a strong consideration for adjuvant treatments, such as thyroid hormone suppression therapy and radioactive iodine. Both of these therapies have been associated with reduced risks of locoregional recurrence and improved disease-free survival in high-risk PTC patients [7]. When nodal status is not known, the role of adjuvant therapy is less clear.pPCND can be used to detect occult central lymph node metastasis and is not only diagnostic, but is also therapeutic and can change management decisions. Due to the results from a pCND, radioactive iodine usage is altered in approximately 30–50% of patients, especially for small tumors [8,24,80]. Even if the decision is to not give radioactive iodine, if a node-positive pathology is determined, complete thyroidectomy and TSH suppression are more strongly considered in lower-risk cases where only a lobectomy has been performed [7].

Furthermore, the implications of surgical de-escalation with respect to locoregional recurrence should not be taken lightly. According to current ATA guidelines, low- or intermediate-risk PTC can be subject to nodal observation without pCND [7]. While clinically apparent preoperative lymphadenopathy has the highest risk of locoregional recurrence with one study estimating a rate of around 22%, the risk of recurrence even in cN0 PTC increases with more involved lymph nodes and the presence of an extranodal extension. A study by Randolph et al. showed that patients with more than five lymph nodes affected had a 19% recurrence risk compared to that of 4% when less than five lymph nodes were involved [81,82]. Without pCND, the ability to accurately assess lymph nodes in the central neck is limited. Not only is pCND diagnostic in cases of occult CNM, but it can be potentially therapeutic. Multiple meta-analyses have shown a statistically significant reduction in locoregional recurrence in patients who have had a pCND compared to those who did not, with Lang et al. citing a 35% risk reduction [10,12,13]. While pCND has been associated with decreased locoregional disease recurrence, its relationship with disease-specific survival is less certain. One study by Barczynski et al. did show increased 10-year disease-specific survival rate in those who underwent a pCND. However, many other papers did not show an improved survival rate, though their mean follow up time was shorter [23,83].

The risks of a pCND should also be considered, especially in primaries with more aggressive features. While some studies have shown no permanent morbidity increases with pCND when compared to those with thyroidectomy alone, multiple meta-analyses have demonstrated an associated increase in hypocalcemia and recurrent laryngeal nerve injury [10,13,14,18] Thus, a risk and benefit discussion regarding pCND and its role in staging and guiding treatment, such as conversion from a lobectomy into total thyroidectomy

with TSH suppression, based on pathologic T staging, should be presented to patients during initial resection, even in small tumors.

This study has several limitations. First, the included studies were primarily retrospective in nature, which may have introduced a selection bias and confounding factors. Second, there was significant heterogeneity in the study populations, different national guidelines, and surgical techniques, which may have impacted the comparability of results and implementation across international populations. The quality of the included studies also varied, with some studies having a lower methodological quality, which could have influenced the overall findings. Finally, in seven studies, the overall CND metastasis rate was not provided and only the ipsilateral versus contralateral metastasis rates were provided [5,6,25,29,33,42,65]. In these cases, the ipsilateral rate was analyzed, which could have led to an underestimation of the metastasis rate.

In conclusion, this study affirms that 30–40% of patients with intrathyroidal tumors measuring <4 cm are understaged, which can impact management decisions. As positive lymph node metastasis, especially that with a high volume of central lymph node involvement (>5), has been associated with increased locoregional recurrence, an ipsilateral pCND can be justified for patients for the more accurate staging and determination of the need for adjuvant treatment [82]. The consideration for pCND is even more important for patients with high-risk genetic mutations such as BRAF V600E. Further research with longer follow up times (see Table 1) is needed to determine the association of pCND with locoregional control by tumor size.

**Supplementary Materials:** The following supporting information can be downloaded at: https://www.mdpi.com/article/10.3390/curroncol30080532/s1. Figure S1: Baujat plots and leave-one-out statistical outputs; Figure S2: Funnel plots and Egger publication bias statistics outputs.

**Author Contributions:** Conceptualization, L.T. and J.C.I.; methodology, L.T., J.P. and R.W.Q.; formal analysis, R.W.Q.; investigation, L.T., J.P. and J.C.I.; writing—original draft preparation, L.T. and R.W.Q.; writing—review and editing, J.P., A.A.S. and J.C.I.; supervision, A.A.S. and J.C.I. All authors have read and agreed to the published version of the manuscript.

**Funding:** This research received no external funding.

**Conflicts of Interest:** The authors declare no conflict of interest.

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
