# Peer review of "Prevalence of Occult Central Lymph Node Metastasis by Tumor Size in Papillary Thyroid Carcinoma: A Systematic Review and Meta-Analysis"

_curroncol, doi:10.3390/curroncol30080532_

Round 1

Reviewer 1 Report

Would you please disccus the impact  of node metastasis in the survival rate . There are metastasis as you showed  but you can discuss better  the impact in survival rate.

Author Response

Thank you very much for your advice. We have now addressed this in lines 267-271

Reviewer 2 Report

In this paper the Authors analayzed the prevalence of Occult Central Lymph Node Metastasis by Tumor Size in Papillary Thyroid Carcinoma. It is a debated and novel topic. A comprehensive and extensive literature review of the NCBI database PubMed was also carried out. The article was well conducted and it is interesting in its fields. It is a well-structured paper, written in good English and the References are up dated. 

Minor issues:

In more demolitive surgery, interventions are affected by more severe complications. In the “discussion” section I suggest to better analyze this topic. Therefore, the following paper should be considered:

“Marotta V, Sciammarella C, Capasso M, Testori A, Pivonello C, Chiofalo MG, Gambardella C, Grasso M, Antonino A, Annunziata A, Macchia PE, Pivonello R, Santini L, Botti G, Losito S, Pezzullo L, Colao A, Faggiano A. Germline Polymorphisms of the VEGF Pathway Predict Recurrence in Nonadvanced Differentiated Thyroid Cancer. J Clin Endocrinol Metab. 2017 Feb 1;102(2):661-671. doi: 10.1210/jc.2016-2555. PMID: 27849428.”

Minor editing of English language required

Author Response

Thank you for your review and guidance. We have addressed your concern in the paper starting in line 273.

Reviewer 3 Report

The aim of this work is a very current topic.
the paper was developed in a linear and coherent manner,
whit a correct language.

  however there are limitations
in addition to those listed
I believe that should be included
at least any genetic mutations and
the associated risk factors.

Author Response

Thank you very much for your guidance. We have added a paragraph in the discussion on genetic mutations and their effects on CMN and recurrence rates (lines 233 – 248)